# New Equations for the Estimation of the Age of the Formation of the Harris Lines

**DOI:** 10.3390/life14040501

**Published:** 2024-04-13

**Authors:** Michał J. Kulus, Kamil Cebulski, Piotr Kmiecik, Patrycja Sputa-Grzegrzółka, Joanna Grzelak, Paweł Dąbrowski

**Affiliations:** 1Division of Ultrastructural Research, Wroclaw Medical University, 50-367 Wrocław, Poland; 2Division of Histology and Embryology, Department of Human Morphology and Embryology, Wroclaw Medical University, 50-367 Wrocław, Poland; kamilcebulski112@gmail.com; 3Institute of Natural and Technical Studies, The Angelus Silesius University of Applied Sciences, 58-300 Wałbrzych, Poland; pkmiecik@ans.edu.pl; 4Division of Anatomy, Department of Human Morphology and Embryology, Wroclaw Medical University, 50-367 Wrocław, Poland; patrycja.sputa-grzegrzolka@umw.edu.pl (P.S.-G.); joanna.grzelak@umw.edu.pl (J.G.); pawel.dabrowski@umw.edu.pl (P.D.)

**Keywords:** growth model, skeletal indicators, growth recovery lines, anthropometrics

## Abstract

Harris Lines (HLs) are transverse, sclerotic lines that can be visualized by X-ray imaging and that occur in long bones, most commonly in the tibia and femur. HLs are associated with disrupted bone mineralization during endochondral ossification, affecting the normal growth process. The etiology of HLs is debated, with some claims linking their presence to detrimental factors such as inflammation, malnutrition, alcohol abuse, and diseases. The age at which HLs form can be estimated based on their location, which allows for a retrospective assessment of the individual’s health status during childhood or youth. The current study is concerned with providing new equations to estimate the age of Harris Line occurrences using a simple calculating tool. Bone growth curves were derived based on a dataset provided by Byers in 1991 using non-linear estimation. The best model was chosen with the Akaike Information Criterion. New and old methods were compared through Bland–Altman plots. As a result, we managed to produce reliable, well-fitted growth curves, concordant with previous methods.

## 1. Introduction

Harris Lines (HLs, also Park–Harris Lines) are transverse, sclerotic lines that can be visualized by X-ray imaging, as shown in Figure 1 [1], although they are visible also in computed tomography [2] or on histological slides [3]. They may form in any bone growing through the process of endochondral ossification. However, they occur most often in the tibia and femur [4]. The first reports of them were in 1874 by Wegner. Later, independent studies conducted by Henry Harris and Edwards Park in the late 1920s described and popularized them [5]. During the last century, HLs were studied intensively in biological anthropology [4], as well as in current radiology [5].

HLs are also referred to as “growth arrest lines” or “growth recovery lines”. According to the model related to the first term, HLs form as a result of disrupted bone mineralization during endochondral ossification. Normally, chondrocytes in the epiphyseal plate proliferate, become hypertrophied, and undergo apoptosis; their remains serve as scaffolds for a new bone matrix produced by osteoblasts. When the proliferation and delivery of chondrocytes to the calcification zone is disrupted, osteoblasts do not receive scaffolding for further bone growth, cannot penetrate the growth plate, and form a thickened layer of bioapatite on an already existing bone [6]. Other models suggest that a thickened layer of bone matrix forms during growth recovery rather than the growth arrest period [7]. Regardless of which model is correct, HL formation is directly linked to transitional bone growth inhibition.

The etiology of HLs remains disputed; their presence has been linked to a broad range of detrimental factors, such as acute inflammation, malnutrition or undernourishment, alcohol abuse, diseases, and others [8,9,10,11]. A recent review by Georgiadis and Gannon [5] provides a list of possible etiologies behind the formation of Harris Lines due to the nature of adverse factors. According to them, physiological, metabolic, endocrine, systemic, post-traumatic, pharmaceutical, and anthropological factors may contribute to their formation.

However, some studies question the reliability of HLs as non-specific detrimental event indicators, which was discussed in a recent review [4]. Furthermore, the assessment of HLs is susceptible to both intra- and inter-observer bias. Some HLs may be missed while, conversely, small cracks may be incorrectly identified as HLs. Differences between observers can be substantial, particularly for faint lines [12]. Intra-observer bias tends to be less significant and is contingent upon the observer’s level of experience. Therefore, it is important to establish clear rules for classifying HLs before beginning measurements to minimize potential discrepancies. Additionally, the use of computed tomography instead of radiographs for assessing HLs results in a slight decrease in inter-observer error [13].

Despite the aforementioned studies questioning the reliability of HLs, they are still used in biological anthropology and paleopathology as non-specific indicators of malnutrition, disease, and other detrimental events [5,14,15,16]. Although bones are remodeled during life, HLs may persist and are detectable for decades, which enables a retrospective evaluation of the health condition of an individual during childhood and adolescence [17]. Thus, the observation of HLs remains a useful tool for paleopathologists, biological anthropologists, and others, but they tend to be considered supportive information rather than standalone proof and are used along with other indicators, such as cribra orbitalia, enamel hypoplasia [16], or traces of physical injury [15].

Among the advantages of evaluating HLs in bioarcheological studies, it should be emphasized that it is an affordable and widely available method, which may be important for low-income countries or for low-cost studies conducted by students or novice scientists.

Besides recording the presence and number of HLs, it is possible to estimate age at their formation [4], which may provide additional information on archeological populations [10,18,19,20]. For example, Hughes et al. [21] noted different distributions of age at HL formation in different age groups, and Jerszyńska and Nowak [18] observed different distributions of HLs in males and females. However, the chronology of the HL formation is rarely calculated despite many available methods. There were 357 articles including the phrase “Harris Lines” published between 2020 and 2023 and indexed by the Google Scholar search engine, yet (to the best of our knowledge) most of them solely recorded the presence of HLs, without their chronology estimation. We managed to find only three recent articles that actually estimated age at HL formation [22,23,24]. Yet there are many articles in which the calculation of HL chronology turned out to be a valuable tool [18,19,20]. Why, then, recently, has the age at HL deposition rarely been taken into consideration?

Besides the controversies mentioned above, the calculation of age at HL formation may still be considered time-consuming and hard to understand and calculate. Six methods exist for estimating age at HL formation: Allison and McHenry’s [25,26]; Hunt and Hatch’s [27]; Clarke’s [28], Maat’s [29]; Hummert and van Gerven’s [30]; and Byers’ [31], recently modified by Kulus and Dąbrowski [4]. Until recently, there were no reviews on how to calculate it and which method may be considered the best one [4]. Research related to determining the chronology of the formation of Harris Lines thus involves a lengthy search for appropriate methods, often described in great detail and requiring a lot of time to learn.

This applies not only to HLs but other indicators; for example, age at linear enamel hypoplasia (LEH) formation may be also calculated with many methods [32,33,34], yet there are a vast number of publications that solely record its presence or lack thereof [35,36]. This may be caused by controversies regarding the precision of LEH chronology estimation [37] or difficulties in calculations and seemingly complex methodologies.

Henriquez and Oxenham [38] recently simplified the calculations of LEH chronology, increasing, concurrently, the precision of age at LEH formation estimation. Their idea was simple yet effective—using existing growth tables, they derived equations describing age at LEH formation as a function of LEH positions on the tooth. Their work simplifies calculations greatly and may help in many further studies on LEH.

The current study is aimed at providing a new tool for fast and precise age at HL formation estimation based on growth curves rather than growth tables using a methodology similar to the one developed by Henriquez and Oxenham for the estimation of LEH chronology. This approach will result in the faster, easier, and more precise calculation of HLs, making them more accessible for scientific use.

## 2. Materials and Methods

### 2.1. Study Population

To evaluate the consistency of the new method, it was used to estimate the age of HL formation in bones from the archeological population of the St. Barbara Church parish cemetery, Wrocław, Poland (Figure 2). The cemetery was established in the 13th century and remained an active burial ground until the 19th century [39]. The remains used for this study were excavated from the area used between the 16th and 18th centuries, which was also confirmed by radiocarbon dating [40].

The cemetery located within the parish of St. Barbara initially served as a burial ground primarily for indigent craftsmen [39]. Later, however, wealthier individuals also found their final resting places here. The necropolis underwent several expansions, continuing until the latter part of the 18th century [41]. The skeletal remains used for the current study were excavated in the latest part of the cemetery and consisted of an early modern population.

The sample consisted of 12 adult tibiae (from 6 males and 6 females) with 137 visible HLs, 6 non-adult tibiae (with 95 visible HLs), and 8 non-adult (age range: 1.5–8 years) femora (with 52 visible HLs). Only bones with abundant HLs were selected for this study.

Bones were imaged with the RTG Quantum Medical Imaging system (SWX RAY, Dallas, TX, USA), using standard parameters (45–50 kV; 17 mA; exposure time, 0.1 s). The total length of the bones and distance from HLs to the nearest bone end were measured using the MicroDicom Viewer (MicroDicom, Sofia, Bulgaria) to the nearest 0.1 cm. Transverse lines extending more than 1/4 the width of the bone were considered valid HLs.

### 2.2. Bone Growth Equation Development and Selection

The method presented in this article is based on Byers [31], which may be considered the best for the estimation of age at HL deposition; among the methods developed to date, this is the most precise. Previous models were discarded because they were based on incorrect assumptions or had significant limitations. The methods of Allison and McHenry [25,26] assumed constant bone growth throughout life. Hunt and Hatch’s method [27] included changes in the bone growth ratio during life; their method was based on precise equations but did not include differences in body (or bone) height between individuals. Clarke’s method [28] did not include epiphyseal thickness, which introduced a constant error. The methods by Maat [29] and Hummert and van Gerven [30] are limited to the distal part of the tibia only.

Byers’ method is free from these disadvantages. It is based on correct and logical assumptions and is not susceptible to errors associated with different bone lengths. It is also suitable for both the distal and proximal parts of the femur and tibia. Further details on the comparison of these methods are presented elsewhere [4]. The principles of Byers’ method and its modification were presented in our latest review [4]; the required measurements are shown in Figure 3.

Briefly, this method requires measurement of (a) the total bone length and (b) the distance from HLs to the closest bone end. Byers [31] developed equations that can determine the ratio of bone length at HL formation to the adult bone length using those two measurements. The calculated ratio may be compared with values in the suitable table to determine age at HL formation. This method was further modified by [4], so it can be used for adult bones and non-adult bones.

Byers’ equations and tables are based on datasets provided by Maresh [42], Gindhart [43], Anderson and Green [44], and Anderson et al. [45]. The data were collected in the mid-20th century (1943–1973) from various institutions in the United States, including the Children’s Hospital in Boston [44,45], the Department of Maternal and Child Health of the Harvard School of Public Health [45], the Fels Research Institute for the Study of Human Development [43], the Child Research Council, and the University of Colorado School of Medicine [42]. It is important to note that the data were collected from white individuals of European descent [43].

The current study required the development of a simple, mathematical model describing the age at LEH formation as a function of the “bone length at HL formation/adult bone length” ratio. Growth curves were developed for tibiae and femora, as these two bone types are used most commonly in the study of HLs [46]. This approach allows for the creation of equations that describe the approximate age at which HL formation occurs. These equations are faster and easier to use than tables that require the calculation of HL chronology to the nearest 0.5 years, as was required in the previous method.

It should be emphasized that using this method does not require an understanding of its basic assumptions. It is enough to look at the diagrams showing the necessary measurements and to follow them.

Growth curves were developed based on the dataset adapted by Byers [31] from the aforementioned studies [42,43,44,45]. Three types of curves were fitted to the mentioned dataset using the non-linear estimation Levenberg–Marquardt algorithm provided by Statistica 13.1 (TIBCO Software Inc., Palo Alto, CA, USA): linear (y = a × x + b), quadratic (y = a × x^2^ + b × x + c), and exponential (y = a×e^(b×x)^ + c). In this equation, y represents the age at HL formation (in years), while x represents the ratio of bone length at HL formation to adult bone length. The parameters a, b, and c are curve-fitting parameters that have been established experimentally.

The optimal model was chosen with the Akaike Information Criterion (AIC), with correction for small samples (AICc) [47,48]. AIC/AICc enables the estimation of the best-fitted and least complicated equations from a set of different models. It helps to find the optimal tradeoff between model complexity and goodness of fit, preventing curve overfitting [49]. Models with the fewest parameters and lowest residuals are scored with the lowest AIC and are considered the best ones.

AIC/AICc were calculated with the AICcmodavg package [50] in the R environment [51]. For the purpose of further calculations, models with the lowest AICc were chosen.

### 2.3. Age at HL Calculation Tool Development

All equations derived in this study are sufficient for standalone calculations of age at HL formation. However, for the convenience of users, they are also provided as ready-to-use formulas, referred to as the ‘age at HL calculation tool’. The age at HL calculation tool was designed as a Microsoft Excel spreadsheet (Appendix A) that uses the equations chosen in Table 1. The spreadsheet automatically calculates the “bone length at HL formation/adult bone length” ratio and approximate age at HL formation. Instructions for the tool are available in the first sheet (entitled “About”).

### 2.4. Bland–Altman Comparison of Method Outputs

To compare the performance of Byers’ method and the method developed in the current study, Bland–Altman plots were used [52]. Each Bland–Altman plot compares the performance of two methods. The X-axis represents the mean value of each pair of results, and the Y-axis represents the difference between the output of the two methods for each pair of measurements. Concordant methods tend to have densely distributed dots and low mean differences. Discrepant methods have more dispersed points and a large mean difference.

## 3. Results

### 3.1. Bone Growth Curve Selection

Table 1 contains bone growth curves based on a dataset provided by Byers [31] for the tibia and femur. Distinct curves were designed for males and females. Moreover, for the calculations of the bones of non-adults, equations based on an averaged dataset were derived since the sex of non-adults (especially infants) cannot be reliably determined [53]. Figure 4 shows curves derived for the tibia.

Each model was scored with AIC and AICc. The model with the lowest AICc was chosen for the age at HL formation calculation tool, although all the models can be used as standalone equations, available for use in any calculation software.

### 3.2. Age at HL Calculation Tool Development

Optimal curves were used to create an age at HL calculation tool, which is available for download as Appendix A. Calculations require bone length [cm], the distance from HL to the closest bone end [cm], the localization of HL (distal or proximal bone part), and sex. The spreadsheet automatically calculates the “bone length at HL formation/adult bone length” ratio (with formulas derived by Byers [4,31]) and approximate age at HL formation (with formulas derived in the current study).

For the bones of children and juveniles, an estimated age of an individual is required. The spreadsheet additionally calculates the ratio of the current bone and “would-be-adult” bone length, which is used for the calculation of approximate age at HL formation.

Detailed instructions are present in the first sheet of Appendix A.

### 3.3. New Method and Byers’ Method Comparison

A comparison of the methods is shown in Figure 5. Bland–Altman plots show great concordance between Byers’ method and the equations derived from the current study—the mean of the difference does not exceed 0.1 years, and the maximum difference does not exceed 0.7 years. The differences between the methods are the result of the imprecision of Byers’ method; his growth tables provide results for full years only, so the estimated result must be rounded to full-year or half-year intervals. Our new method is based on curve equations and does not have such limitations.

## 4. Discussion

In biological anthropology (and related fields), vast amounts of data may be ignored just because of the difficulties in calculations. Providing new, fast, and accurate tools may increase the quality of future studies. Although the method developed by Byers is still applicable, it has one major disadvantage—the tables provided for age at HL deposition present values for full years only; therefore, the final results must be given in one-year or half-year intervals. Rounding the age at which HLs form can lead to decreased accuracy, particularly for HLs that are so close together that they must be classified as forming at the same time, which is an obvious error.

Moreover, without the latest modification [4], Byers’ method could only be used for adult bones. Methods based on equations rather than tables are more accurate and less subjective [38]. Moreover, they enable faster calculations and may be easily used after basic training. These derived and selected equations fit the bone growth datasets accurately and enable calculations for adult and non-adult bones.

The previous, modified Byers’ method [4] was tested for consistency using an artificial model of bones with abundant HLs. For a change, the method developed in this study was evaluated using actual bones. Although the population was rather limited, the equations derived in the current study show great concordance with the previous method, with differences not exceeding 0.7 years. It may be assumed that both methods are highly concordant and valid, but the newer method allows for increased estimation accuracy.

The method developed in this study is intended for use in biological anthropology and related fields, although it is not limited to them. Observing Harris Lines has clinical significance [5] and may be used to estimate age at HL formation in contemporary populations—the dataset used for the development of the current method was obtained in the 20th century [31] and shows similar growth patterns to modern populations.

Importantly, the application of this method does not require a prior understanding of its underlying assumptions. Users can achieve successful implementation by following the provided diagrams, which clearly illustrate the required measurements, and then using appropriate equations or Appendix A. However, there is room for the optimization of this method. The modification that yields the best results, albeit the most challenging, is the source dataset for bone growth. It is important to note that different populations may exhibit variations in bone growth patterns, particularly during the adolescent growth spurt [54,55,56]. Creating our own growth curves based on the study population would result in the most accurate estimation of age at the formation of HLs. This is particularly important for populations that differ significantly from the mid-20th century white North American population.

A future tool for calculating age at HL formation may allow the equations to be customized based on the growth patterns of the population being studied. Assuming the same tibia (or femur)/body height ratio, based on decile charts for distinct populations, it may be quite possible to create customized equations for them. Although this would require more sophisticated tools than those used for the current study, it may be considered a future direction in the calculation of age at HL formation.

HLs cannot be correlated with bone length and morphology [1,57] or age at linear enamel hypoplasia (LEH) formation [58,59]. The lack of correlation with LEH may be considered a contradiction since both HLs and LEH possibly share similar etiologies [33,58]. However, this may be logically explained: LEH usually forms on anterior teeth, which are formed during early childhood [60]. HLs may form during the whole bone growth period; however, early HLs tend to disappear during the bone-remodeling process [58]. The evaluation of LEH and HLs in juvenile or infant individuals, who tend to have more pronounced and visible HLs [30], could lead to finding some correlation between both indicators. However, to the best of our knowledge, no such study has been conducted. Moreover, the reason may also be linked to the calculation methods of age at LEH/HL formation.

Until recently, the most accurate method of LEH formation estimation was the decile growth chart-based method developed by Reid and Dean [32,61]. The equation-based method by Goodman and Rose was deemed too simplistic to properly describe enamel growth [38,62]. Cumulative inaccuracies may impede the correlation of both indicators. A study on the remains of juvenile individuals, utilizing the Henriquez and Oxenham method for age at LEH formation estimation and the currently developed method for age at HL formation estimation could bring more insights into the correlation (or lack thereof) between the two indicators.

It is important to note that the growth curves derived in this study are not universal. Byers’ bone growth tables were developed based on a study of the North American white population in the mid-20th century [42,43,44,45], so they may be unsuitable for populations with different growth patterns. To achieve the most accurate estimation of HL chronology, new bone growth tables and models should be developed for the population under study following the methods presented in this study.

Also, this method shares inevitable disadvantages with the previous methods, as described in our previous review [4]. This method cannot take into consideration individual variations in bone growth, such as delayed or premature adolescent spurts, developmental defects, stunted growth, and others. Moreover, when evaluating age at HL formation in non-adult individuals, the possible inaccuracy of age estimation further decreases the precision of age at HL estimation. Therefore, it is important to evaluate age at HL formation in fairly large populations of study, to blur the influence of individual variability, and to include the abovementioned sources of inaccuracies in the limitations of studies.

## 5. Conclusions

This study introduces a novel equation-based method of estimating the age at which Harris Lines (HLs) form in bones, presenting significant improvements over previous approaches. An evaluation of its consistency using actual bones demonstrates high concordance with the existing methodologies but also higher accuracy. The new method paves the way for fast, accurate, and reliable age at HL formation estimation in biological anthropology and related fields.

## Figures and Tables

**Figure 1 life-14-00501-f001:**
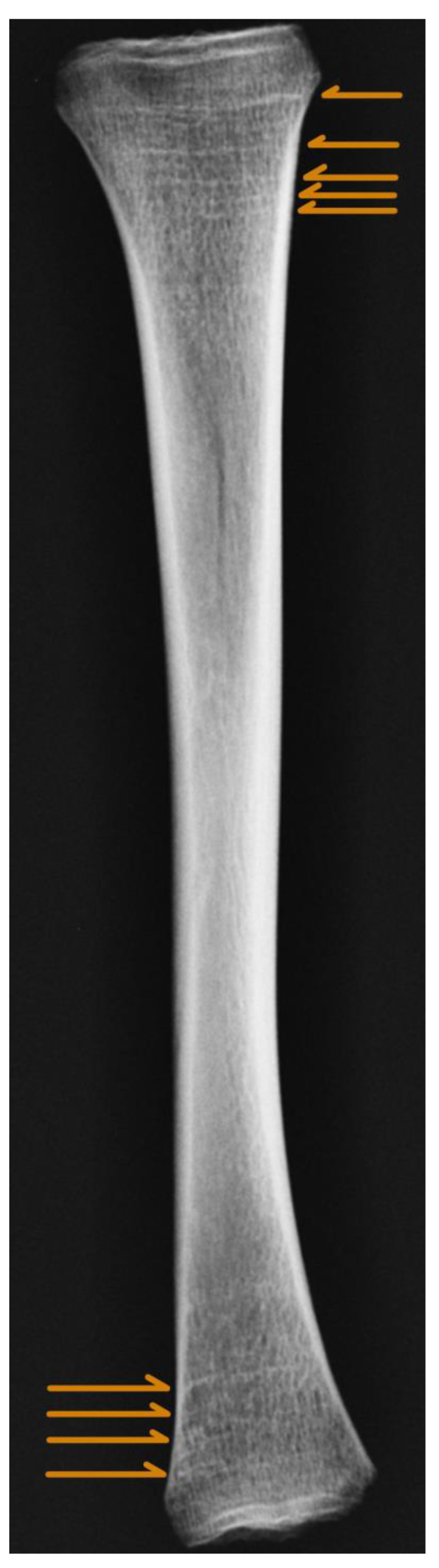
Example of Harris Lines (orange arrows) on juvenile tibia. Source: own archive.

**Figure 2 life-14-00501-f002:**
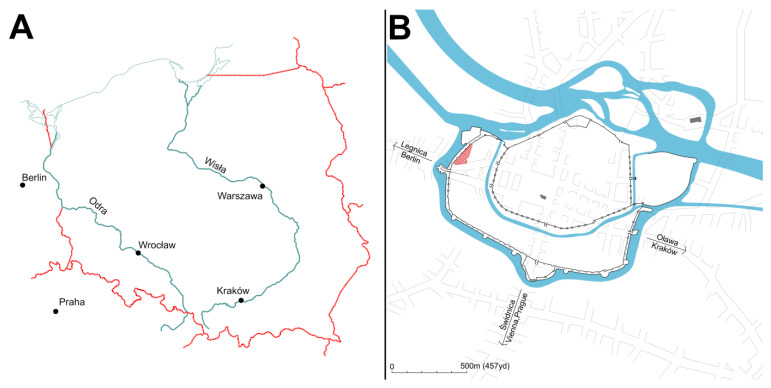
(**A**) Localization of Wrocław, Poland, and (**B**) cemetery of St. Barbara Church (red area).

**Figure 3 life-14-00501-f003:**
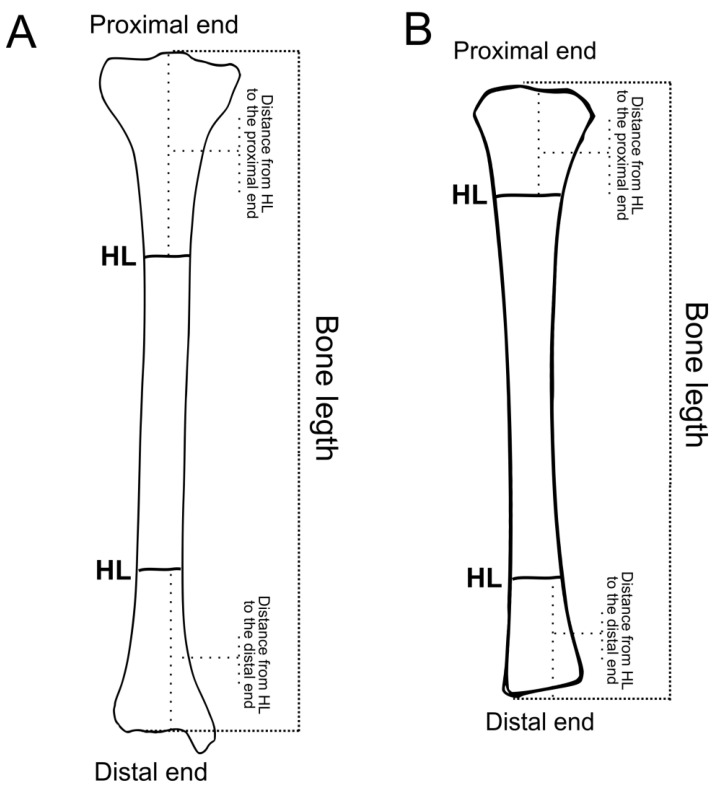
Measurements required for calculation of age at HL deposition with Byers’ method. Calculations require two measurements: (a) total bone length and (b) distance from the HL to the nearest bone end. The method’s principle is described in a recent open-access review [4]. (**A**) adult bone; (**B**) bone of non-adult.

**Figure 4 life-14-00501-f004:**
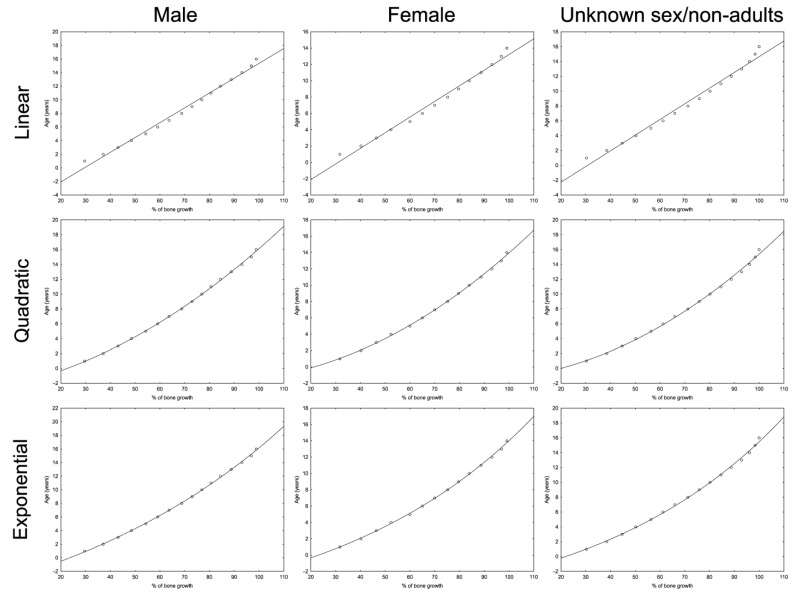
Bone growth curves. Dots indicate the values from bone growth tables provided by Byers [31]. Equations for each curve are shown in Table 1.

**Figure 5 life-14-00501-f005:**
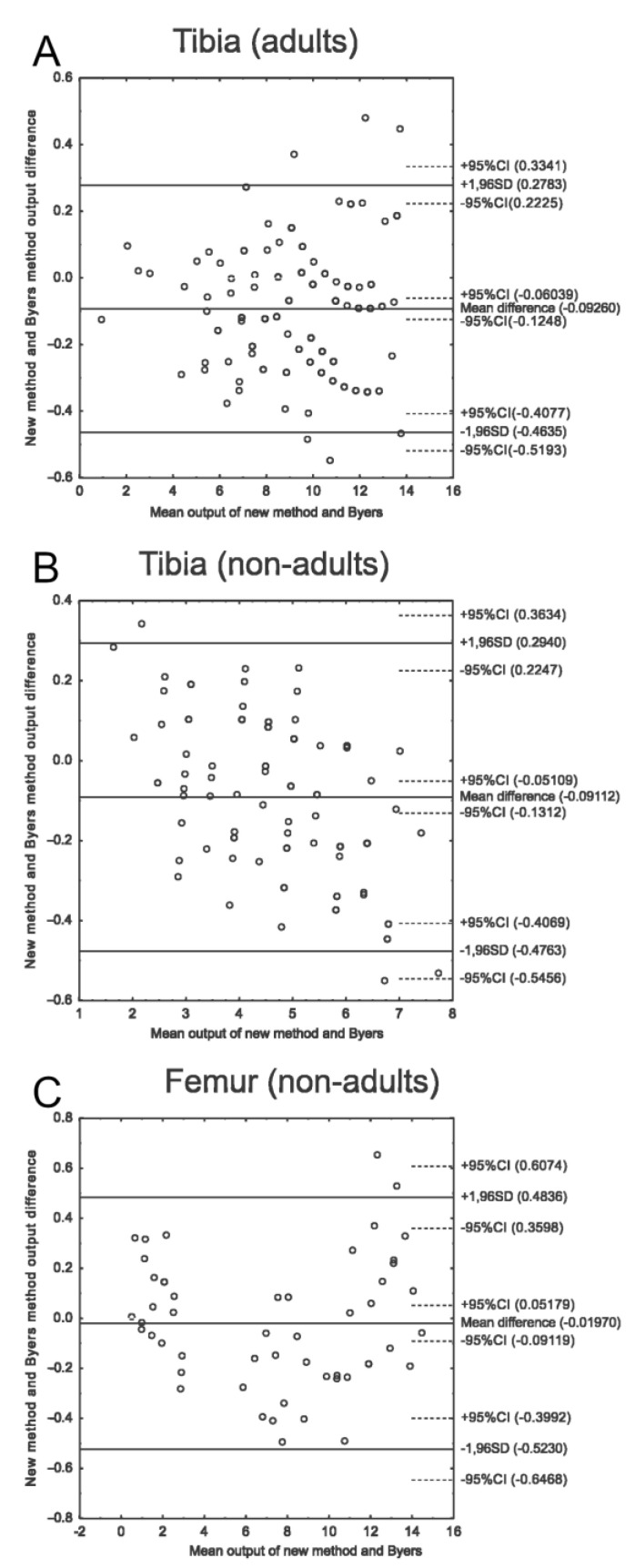
Bland–Altman plot comparing Byers’ method and the new equations. The X-axis represents the mean output of two methods for one HL; the Y-axis represents the difference between the method outputs for each HL. The mean difference for tibias (**A**,**B**) is low and does not exceed 0.5 years; slightly worse results were obtained for femurs (**C**).

**Table 1 life-14-00501-t001:** Choosing the optimal equation. The *y* stands for age (given in years); *x* stands for the “bone length at HL formation/adult bone length” ratio (given in percentages); The lowest AIC indicates the optimal curve; AICc represents the AIC with correction for small groups. Residuals represent the sum of squared differences between the model and source values. Equations with the lowest AICc are bold.

Bone	Equation Type	Equation	AIC	AICc	Residuals
Femur (males)	Linear	*y* = 0.217859 × *x* − 6.43422	25.82	27.82	3.234
	Quadratic	** *y* ** **= 0.00106893 × *x*^2^ + 0.077086 × *x* − 2.28129**	**−18.35**	**−14.71**	**0.1804**
	Exponential	*y* = 11.5213 × e^0.00979342×*x*^ – 14.5196	−15.94	−12.30	0.2098
Femur (females)	Linear	*y* = 0.191588 × *x* − 5.93988	24.19	26.59	3.007
	Quadratic	*y* = 0.00114036 × *x*^2^ + 0.0387797 × *x* − 1.33028	−12.01	−7.56	0.1963
	Exponential	** *y* ** **= 6.84604 × e^0.0121551×*x*^ − 9.07047**	**−14.22**	**−9.77**	**0.1676**
Tibia (males)	Linear	*y* = 0.216145 × *x* − 6.14786	20.56	22.56	2.328
	Quadratic	** *y* ** **= 0.000860092 × *x*^2^ + 0.103426 × *x* − 2.84765**	**−13.35**	**−9.71**	**0.2467**
	Exponential	*y* = 16.227 × e^0.00787971×*x*^ − 19.5643	−11.19	−7.55	0.2823
Tibia (females)	Linear	*y* = 0.19219 × *x* − 5.90254	22.61	25.01	2.685
	Quadratic	*y* = 0.00105703 × *x*^2^+ 0.0508895 × *x* − 1.64875	−4.59	−0.15	0.3333
	Exponential	** *y* ** **= 7.78153 × e^0.011363×*x*^ −10.1631**	**6.38**	**−1.94**	**0.2933**
Tibia (unisex)	Liner	*y* = (0.211442) × *x* + (−6.50326)	33.78	35.78	5.318
	Quadratic	*y* = 0.00128889 × *x*^2^ + (0.0374516) × *x* + (−1.24991)	4.29	7.93	0.7431
	Exponential	** *y* ** **= (6.79694) × exp^0.0128072×*x*^ + (−9.00042)**	**1.11**	**4.75**	**0.6093**
Femur (unisex)	Linear	*y* = (0.211759) × *x* + (−6.65994)	36.57	38.57	6.331
	Quadratic	*y* = (0.00142038) × *x*^2^ + (0.0191624) × *x* + (−0.812149)	6.1	9.74	0.8323
	Exponential	** *y* ** **= (5.61797) × exp^0.0140969×*x*^ + (−7.60929)**	**2.58**	**6.21**	**0.6677**

## Data Availability

The data are contained within the article and Appendix A.

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
