# Peer review of "New Equations for the Estimation of the Age of the Formation of the Harris Lines"

_life, 2024, doi:10.3390/life14040501_

Round 1

Reviewer 1 Report

Comments and Suggestions for Authors

The authors present an innovative equation-based approach for estimating the age at which Harris lines develop on bones. This novel method offers several advantages compared to existing techniques. The inclusion of supplementary material alongside the method's description facilitates its straightforward application. Consequently, I find the manuscript to be of significance and deserving of publication, although it necessitates small corrections. Attached is a non-comprehensive list of identified issues and suggestions for enhancement.

Comments on the Quality of English Language

The English language is suitable, requiring only minor corrections.

Author Response

Thank you for your review and detailed proofreading – it helped a lot. We've made all the corrections you mentioned in your review. Typos and references have been corrected. We've removed some redundant information from Table 1 and moved part of the description of Figure 5 to the Results section. Also, there's no typo in Table 1 - I double-checked my calculations, the AIC value mentioned is a positive number, but the AICc value was the most important in the model selection.

Reviewer 2 Report

Comments and Suggestions for Authors

Dear Editor of Life, and Authors,

I have read the submited manuscript and I have found it relevant for the fields of study of bioarcheology and paleopathology, trying to address a technical flaw related with the estimation of age at which Harris lines appear in long bones. The articles makes a good referencing of previous work, and engenders a compelling argument about the relevance of the study of Harris lines in human skeletal remains from the past. The construction of an Excel tool to facilitate the calculations is also commendable. However, in order to be published in Life, the manuscript needs some amendments:

1. The data used to generate the formulae for estimate the age at which the growth arrest lines form is supposedly derived from Byers (1991) but, in fact, it is from the database provided by Maresh (1955), and also data from Anderson et al., 1963; Anderson and Green, 1948; and Gindhart, 1973. Thus, data used needs to be described in a more meaningful way, addressing the problems associated with it (Maresh's data, for example, comes from a middle class, euro american, background) and defining it in terms of geographic origin, chronology, etc. A more thorugh description of the original data (not derived from Byers!) should appear in the MAterials and Methods section.

2. The method was not "validated" in the proper sense, but it showed an internal consistency with the Byers method. Only if the authors had access to the real age at which the lines appeared they could make a validation study. As such, the focus must be on the internal consistency results between two ESTIMATION methods. This is a very important conceptual difference, and needs to be addressed in the discussion. So the conclusion - "Validation against actual bones demonstrates high concordance with existing methodologies, but higher accuracy" - is not correctly addressed, as accuracy (the difference between estimated age of appearance and the real age of appearance) is impossible to calculate with the empirical data used in the study. Only internal consistency between methods can be estimated.

3. As the method is based on estimations upon estimations (age must also be estimated in non-adults for the method), there is much more uncertainty attached to it, and this must be clearly highlighted in the limitations of the method.

4. Replace "physical anthropolgy" with "biological anthropology".

My best regards.

Comments on the Quality of English Language

Minor amendments required.

Author Response

Thanks for the review - you made good points and raised important issues with which we could not disagree. The exact sources on which Byers' method (and our modification) is based were mentioned in our previous review ("How to calculate the age at formation of Harris lines? A step-by-step review of current methods and a proposal for modifications to Byers' formulas", 2019), but we completely agree that it should have been mentioned in the current article. We've added an appropriate paragraph in the Materials and Methods section, as well as mentioning the original source of the base dataset in the Discussion. This is important because it shows the main (though unavoidable) limitation of the method, which is the possible difference in growth pattern between different populations - which was additionally underlined in the Discussion section.

The issue of "double estimation" is also addressed in the last part of the discussion - we've added two sentences highlighting the problem of evaluating non-adult bones.

We agree with your point about the inappropriate use of the term "validation". We've replaced that word with more appropriate terms, such as "internal consistency," as you suggested. The term "physical anthropology" has been replaced with "biological anthropology".

Reviewer 3 Report

Comments and Suggestions for Authors

The paper is very interesting, clear and bring value to the field. I have only, let's say, technical advises related to the way the information is presented. I checked the supplementary files provided and they just simply apply the equations on the main paper and describe the process. I suggest to skip using supplementary files since, for instance, for printed formats, either the excel or the movie do not bring more values than the paper already does and even more, they cannot be printed (well, the Excel might but still, not very comfortable). The target readers are supposed to have at least a minimum level of knowledge on the computing field (I guess) so they should be able to apply the valuable information from the article to reproduce the results you obtained. At the end of the day, this is actually one important request from a scientific paper, meaning to be easy readable and reproductible so the others to be able to check at least a minimum level of the knowledge from the paper and make sure they get the value. I suppose that technical details on Excel spreadsheets and training movies should remain between you and others interested to speed up the familiarization with the topic you approached, using, most probably, the email of the corresponding author. I guess you should better make a note about that deeper technical details might be made available upon request. 

I hope that I made my point. I don't want to be mean, I just try to keep the paper focus on the scientific information and not on the pen used to write the paper. Bellow I let you know what I would change in the article. I hope I didn't miss any other lines.

At the lines:

- 201-202 - remove "(Supplementary File 1)"

- 204-205 - remove "Instructions for the tool are available in the first sheet (entitled “About”) and the 1-minute videotutorial is available as Supplementary File 2."

- 238-239 - remove "tool, which is available to download as Supplementary File 1"

- 241 - replace Spreadsheet automatically calculates with something else such as "The calculation tool automatically calculates ..."

- 289-291 - rephrase " ... and then inserting them into the appropriate cells of the spreadsheet provided as Supplementary Material 1. The video tutorial provided would make it easy to learn how to use this calculator even without reading the current manuscript." with something more general such as "Using the calculation tool, the results are automatically provided" ... I mean, this is not a training material on how to use Excel tool or other similar tool. Please focus the paper on the main topic you picked up, that is very interesting.

- 342-344 - please remove Supplementary Materials: S1: Age at HL formation calculation tool. An Microsoft Excel spreadsheet facilitating calculation of HLs chronology. All instructions are included in the first sheet (entitled: “About”). S2: 1-minute long videotutorial presenting how to use Supplementary File 1.

I would rather add a new sub-chapter called Technical steps and I would describe the spreadsheet with some (2-3) images and few steps what the spreadsheet does. The movie, from my standpoint, is not appropriate.

Comments on the Quality of English Language

At the line 35, the phrase "They may be formed in any bone growing through the process of endochondral ossification, however they occur most often in tibia and femur [4] ..." should be split in "They may be formed in any bone growing through the process of endochondral ossification. However, they occur most often in tibia and femur [4]."

At the line 150, please correct the thourought with thourough.

At the line 192, correct ofthe with of the.

At the line 194, correct fit,preventing with fit, preventing (add a space after comma).

Author Response

Thanks for your insightful review. I can clearly see your point and agree with it to a certain extent. In particular, I agree that the focus should be shifted slightly from the calculation tool to the equations and their derivation - that's why we decided to follow your suggestion and remove Supplementary File 2, as well as to underline in some parts of the manuscript that the equations derived in the current study can be used stand-alone, without the calculation spreadsheet we developed. Also, as you suggested, we decided to include "technical steps" instead of a video tutorial. They are now included as a graphical abstract.

Nevertheless, we've decided to leave Supplementary File 1 in this article, and here's why we think it should be included. As you wrote: “The target readers are supposed to have at least a minimum level of knowledge on the computing field (I guess) so they should be able to apply the valuable information from the article to reproduce the results you obtained”; this assumption would be true if this article were addressed only to computer scientists or bioinformaticians. But it's addressed also to potential users – biological anthropologists, radiologists, paleopathologists. And based on what I've learned from interacting with them, the abovementioned assumption is somewhat optimistic. I respect their knowledge and expertise, usually they are proficient in many fields and basic use of computers is common among them, but for many of them even a simple IF/ELSE formula is a sophisticated concept and implementing our equations in even such a basic tool as MS Excel would be difficult (or even discouraging) for them.

That's why any tool or concept that can help them make potentially useful calculations is necessary. For example, Henriquez and Oxenham in their "New distance-based exponential regression method and equations for estimating the chronology of linear enamel hypoplasia (LEH) defects on the anterior dentition (2019)" provided a simple spreadsheet with even simpler formulas. But in 5 years they were cited 38 times (according to Google Scholar), which shows how necessary such tools are for the scientific community.

Moreover, as we noted in the introduction, HLs have been the subject of hundreds of studies, yet the calculation of their age of formation is currently rarely used - most likely because it's considered difficult. Our previous work ("How to calculate the age at formation of Harris lines? A step-by-step review of current methods and a proposal for modifications to Byers' formulas", 2019) was aimed at facilitating the calculations, but it turned out that we need to provide further support to biological anthropologists in this area.

I am aware that the calculation tool in the Excel spreadsheet is as simple as a high school project, but I had to keep the end users in mind. I also considered more "professional" tools, such as creating the R statistical environment package - but that would dramatically reduce the pool of potential users and wouldn't solve the problem, so I decided to stick to the famous "KISS" (Keep it simple, straight) concept and use the program that more users are familiar with.

To be clear, I really appreciate your feedback, as it was thoughtful and forced me to think about important issues related to the current article – and future studies as well. I hope that even though I couldn't follow all of your suggestions, you will find my rationale satisfactory.

Round 2

Reviewer 2 Report

Comments and Suggestions for Authors

Dear Editor of Life and Authors,

I believe that all the problems that I have identified in the original manuscript were well corrected and the second revision is much improved.

Best regards.